# Open Software/Hardware Platform for Human-Computer Interface Based on Electrooculography (EOG) Signal Classification

**DOI:** 10.3390/s20092443

**Published:** 2020-04-25

**Authors:** Jayro Martínez-Cerveró, Majid Khalili Ardali, Andres Jaramillo-Gonzalez, Shizhe Wu, Alessandro Tonin, Niels Birbaumer, Ujwal Chaudhary

**Affiliations:** 1Institute of Medical Psychology and Behavioural Neurobiology, University of Tübingen, Silcherstraße 5, 72076 Tübingen, Germany; 2Wyss-Center for Bio- and Neuro-Engineering, Chemin des Mines 9, Ch 1202 Geneva, Switzerland

**Keywords:** electrooculography (EOG), Human-Computer Interface (HCI), Support Vector Machine (SVM)

## Abstract

Electrooculography (EOG) signals have been widely used in Human-Computer Interfaces (HCI). The HCI systems proposed in the literature make use of self-designed or closed environments, which restrict the number of potential users and applications. Here, we present a system for classifying four directions of eye movements employing EOG signals. The system is based on open source ecosystems, the Raspberry Pi single-board computer, the OpenBCI biosignal acquisition device, and an open-source python library. The designed system provides a cheap, compact, and easy to carry system that can be replicated or modified. We used Maximum, Minimum, and Median trial values as features to create a Support Vector Machine (SVM) classifier. A mean of 90% accuracy was obtained from 7 out of 10 subjects for online classification of Up, Down, Left, and Right movements. This classification system can be used as an input for an HCI, i.e., for assisted communication in paralyzed people.

## 1. Introduction

In the past few years, we have seen an exponential growth in the development of Human-Computer Interface (HCI) systems. These systems have been applied for a wide range of purposes like controlling a computer cursor [1], a virtual keyboard [2], a prosthesis [3], or a wheelchair [4,5,6,7]. They could also be used for patient rehabilitation and communication [8,9,10,11]. HCI systems can make use of different input signals such as voice [7], electromyography (EMG) [12], electroencephalography (EEG) [13], near-infrared spectroscopy (NIRS) [14,15,16] or electrooculography (EOG) [5].

In this paper, we describe an EOG classification system capable of accurately and consistently classifying Up, Down, Left, and Right eye movements. The system is small, easy to carry, with considerable autonomy, and economical. It was developed using open hardware and software, not only because of economic reasons, but also to ensure that the system could reach as many people as possible and could be improved and adapted in the future by anyone with the required skills. 

The end goal of this work is to build a system that could be easily connected to a communication or movement assistance device like a wheelchair, any kind of speller application, or merely a computer mouse and a virtual keyboard.

To achieve these objectives, we have developed and integrated the code needed for:Acquiring the Electrooculography (EOG) signals.Processing these signals.Extracting the signal features.Classifying the features previously extracted.

EOG measures the dipole direction changes of the eyeball, with the positive pole in the front [17]. The technique of recording these potentials was introduced for diagnostic purposes in the 1930s by R. Jung [18]. The presence of electrically active nerves in the posterior part of the eyeball, where the retina is placed, and the front part, mainly the cornea, creates the difference in potential on which EOG is based [19]. This creates an electrical dipole between the cornea and the retina, and its movements generates the potential differences that we can record in an EOG. 

There are several EOG HCI solutions present in the literature. One of the issues with current HCI systems is their size and lack of autonomy, the use of proprietary software, or being based on self-designed acquisition and processing devices. Regarding the acquisition system, the most common approach is to use a self-designed acquisition device [1,4,20,21,22]. In our view, this solution dramatically restricts the number of users who can adopt this system. Other proposed systems make use of commercial amplifiers [23,24], which in turn make use of proprietary software and require robust processing systems, mainly laptops. This also reduces the number of potential users of the system and its applications since it increases the cost of the system and reduces its flexibility, portability, and autonomy. As far as signal processing is concerned, most systems choose to use a laptop to carry out these calculations [1,20,21,22,24,25], but we can also find the use of self-designed boards [6,26]. Table 1 shows the characteristics of some solutions present in literature as a representation of the current state of the art. The goal of our work is to achieve results equivalent to the present state of the art using an open paradigm, demonstrating that it is possible to arrive at a solution using cheaper components that could be modified to build a tailored solution. As far as we know, this is the first time that an open system is presented in this scope.

In our system, the signal is acquired using the OpenBCI Cyton Board (Raspberry Pi 3B+ official website), a low-cost open software/hardware biosensing device, resulting in an open hardware/software-based system that is portable, with considerable autonomy and flexibility. 

Once we have the EOG signal, this is processed using a Raspberry Pi (OpenBCI Cyton official website), a single board computer that allows installing a Linux-based distribution, which is small, cheap, and gives us the option to use non-proprietary software.

Features are then extracted from the acquired signal and classified employing a machine learning algorithm. The feature extraction process aims to reduce the dimensionality of the input data without losing relevant information for classification [28] and maximizing the separation between elements of different classes by minimizing it between elements of the same class [27]. To achieve this, several models have been proposed on EOG feature extraction [29,30,31,32]. We employed Support Vector Machine (SVM) to classify the data [33,34], which creates a boundary to split the given data points into two different groups.

The result of this process, in the context of signal (EOG) mentioned in this article, is the classification of the subject’s eye movement to be used as input commands for further systems. This process and the tools used for it are explained in detail in Section 2. The Section 3 shows the performance achieved by the system. Finally, in the Section 4, we discuss the designed system and compare our system with existing related work along with the limitations of our system and future work.

## 2. Materials and Methods

### 2.1. Hardware-Software Integration

In the present study, OpenBCI Cyton board was used for the signal acquisition. This board contains a PIC32MX250F128B microcontroller, a Texas Instruments ADS1299 analog/digital converter, a signal amplifier and an eight-channel neural interface. This device is distributed by OpenBCI (USA). Figure 1 depicts the layout of the system.

This device gives us enough precision and sampling rate (250 Hz) for our needs, it has an open-source environment (including a Python library to work with the boards (OpenBCI Python repository)), it has an active and large community^,^ and it can be powered with a power bank, which is a light and mobile solution. Attached to the board, we have 4 wet electrodes connected to two channels on the board in a differential mode. Differential mode computes the voltage difference between the two electrodes connected to the channel and doesn’t need a reference electrode. The two channels correspond to the horizontal and vertical components of the signal.

The acquisition board is connected to a Raspberry Pi, a single-board computer developed by the Raspberry Pi company based in the United Kingdom. Although its firmware is not open source, it allows installing a Linux-based distribution keeping the open paradigm in our system. In this case, we chose to install Raspbian, a Debian-based distribution. The hardware connection between the OpenBCI board and the Raspberry Pi is made using a wireless RFDuino USB dongle. On the software side, we used an open Python library released by OpenBCI. To run this library over the Raspberry Pi, the source code of the mentioned library has been partially modified. It has also been necessary to recompile some third-party libraries so that they could run on the Raspberry Pi. We decided to power both the OpenBCI board and the Raspberry Pi via a USB connection to a power bank (20,000 mAh) to maximize the system autonomy and mobility.

This hardware configuration offers us all the characteristics that we were looking for: it has enough computational power to carry our calculations, it’s small and light, it allows us to use free and open-source software, and is economical. It should be noted that although we have used the OpenBCI board as acquisition system there are some other solutions that fit our needs like the BITalino biosignal acquisition board. This board offers an EOG acquisition module and an open environment which includes a Python-based API for connection and signal acquisition over Raspberry Pi.

It should be mentioned that the data presented in this article have been processed using a conventional laptop instead of the Raspberry Pi, just for the convenience of the experimenters. During the development of the research, several tests were carried out that did not show any difference in the data or the results depending on the platform used.

We decided to use EOG over other eye movement detection techniques like Infrared Reflection Oculography (IROG) [35] or video-based systems [25], as the EOG technique does not require the placement of any device that could obstruct the subjects’ visual field. Four electrodes were placed in contact with the skin close to the eyes to record both the horizontal and the vertical components of the eye movements [36,37].

### 2.2. Experimental Paradigm

Ten healthy subjects between 24 and 35 years old participated in the study and gave their informed consent for inclusion. The signal acquisition was performed in two stages: training and online prediction. For both stages, we asked the subjects to perform four different movements: Up, Down, Left, and Right. Each movement should start with the subject looking forward and then look at one of these four points already mentioned and look again at the center. For the training stage, we acquired two blocks of 20 trials, 5 trials per movement. In these blocks, five “beep” tones were presented to the subject at the beginning of each block in 3 s intervals to indicate the subject the interval that they had to perform the requested action. After these initial tones, the desired action was presented via audio, and a “beep” tone was presented as a cue to perform the action. The system recorded during the 3 s after this tone was presented, and the system presented again another action to be performed. For some of the subjects, these two training blocks were appended in a single data file. The schematic of the training paradigm (offline acquisition) is shown in Figure 2a.

The online classification was performed with a block of 40 trials, 10 per movement, on Subject 1. After this experiment, we decided to reduce the number of trials per block to 20, 5 per movement, for the convenience of the subject. This online block had the same characteristics as the classification blocks except that the five initial tones were not presented, and the actions to be performed were separated by 5 s interval to have enough time for the prediction tasks. Furthermore, in these blocks, the system recorded only during the 3 s after the cue tone was presented. During this stage, we generate two auxiliary files: one with the acquired data and the other containing the action that the user should perform and the action predicted. We only considered predicted actions with a prediction probability higher than a certain threshold. For the first subject, we set this threshold as 0.7, but after that experiment, we changed the threshold to 0.5. In this case, the auxiliary file corresponding to subject 1 contains the predictions made using 0.7 as a prediction probability threshold. Figure 2b depicts the schematic of the online prediction paradigm.

### 2.3. Signal Processing

A second-order 20 Hz lowpass Butterworth filer [37] was used to remove the artifacts arising from electrodes or head movements and illumination changes [19,27,38]. A 20 Hz lowpass filter was used because the artifacts, as mentioned earlier, appear in the high frequencies [17], and the EOG signal information is contained mainly in low frequencies [30]. The irregularities in the signal after the lowpass filter were removed using a smoothing filter [30]. For applying these filters, we used the SciPy library. This library is commonly used and has a big community supporting it. 

The last step in pre-processing was to standardize the data. This is done to remove the baseline of EOG signals [27]. The standardization was done using the following formula:(1)$$X_{t}=\frac{x_{t}-\;\mu_{i}}{\sigma_{i}},$$
where *i* is the sample that we are processing, *t* corresponds to a single datapoint inside a sample, $X_{t}$ is the resulting datapoint, $x_{t}$ is the data point value before standardization, $\mu_{i}$ is the mean value of the whole sample and $\sigma_{i}$ is the standard deviation of the whole sample. An example of the processed signal can be seen in Figure 3, which shows a single Down trial extracted from a classification block of Subject 5.

Figure 4 depicts the vertical and horizontal component for four different eye movement tasks performed by subject 5.

### 2.4. Feature Extraction

An essential step in our system’s signal processing pipeline is feature extraction, which for each sample, calculates specific characteristics that will allow us to maximize the distance between elements in different classes and the similarity between those that belong to the same category. We use a model based on the calculation of 3 features for the horizontal and vertical components of our signal, i.e., 6 total features per sample. The features are the following:Min: The minimum amplitude value during the eye movement.Max: The maximum amplitude value during the eye movement.Median: The amplitude value during the eye movement that has 50% values above as below.

### 2.5. Classification

Once we have calculated the features of each sample, we create a model using that feature values and its class labels. Even though some biosignal-based HCI use other machine learning techniques, such as artificial neural networks [29,36] or other statistical techniques [19], most of the HCI present in the literature use the machine learning technique called Support Vector Machine. We have decided to use SVM because of its simplicity over other techniques, which results in a lower computational cost and excellent performance. 

In this study, we have used the implementation of the SVM of Scikit-Learn, a free and open-source Machine Learning Python library. This library has a high reputation in Machine Learning, and it has been widely used. The selected parameters for creating the model are a Radial Basis Function (RBF) as kernel [39], which allows us to create a model using data points that are not linearly separable [40], and a One vs. One strategy [41], i.e., creating a classifier for each pair of movement classes. Finally, we have performed 5-fold cross-validation [42], splitting the training dataset into 5 mutually exclusive subsets and also creating 5 models, each one using one of these subsets to test the model and the other four to create it. Our model accuracy is calculated as the mean of these 5 models.

## 3. Results

The acquired signal is processed to remove those signal components that contain no information, resulting in a clearer signal. The data were acquired from 10 healthy subjects between 24 and 35 years old. The result of signal processing can be seen in Figure 3 and Figure 4, which shows the single trials of a training block performed by Subject 5. As Figure 3 and Figure 4 show, the result of this step is the one expected. For Subject 8, we found flat or poor-quality signals in the vertical and horizontal component, so we decided to stop the acquisition and discard these data. Some trials extracted from this discarded block can be seen in Figure 5 which shows no clear steps or any other patterns for the four movements. This situation is probably due to an electrode movement, detachment, or misplacement that could not be solved during the experiment.

After artifact removal, feature extraction is performed to reduce the dimensionality in input, leading to characteristics that define the signal without information loss. As mentioned above, the features used were Maximum, Minimum, and Median. It should be noticed that Up and Down movements have relevant information only for the vertical channel of our signal as well as Left and Right movements have this relevant information in the horizontal component. Figure 6 and Figure 7 present an example of this feature extraction process over two blocks of 20 trials, each corresponding to the training data of Subject 5, who ended up with 100% accuracy. Figure 8 and Figure 9 present an example of the same feature extraction process over two blocks of 20 trials performed by Subject 6, who ended up with 78.7% accuracy. In these figures, we can appreciate that Subject 5, with 100% accuracy, shows a more evident difference in the data values than Subject 6, with 78.7% accuracy. Figure 8 and Figure 9 show some overlapping in the data values, which explains the lower classification accuracy achieved.

The last step in our pipeline is to build a model and perform an online classification of the subject’s eye movements. As we mentioned before, we build our model using 5-fold cross-validation. Table 2 shows the model accuracy, the accuracy-related on how good the model has been classifying the training data, as the mean of these five models for each subject. For the prediction accuracy—the accuracy related to the prediction of unseen data—we have asked the subject to perform 20 movements per block (five of each movement), as is explained in Section 2.2. We predicted those movements using the pre-built model and, finally, validated how accurate that prediction was.

As mentioned in Section 2.2., we only consider those predicted actions with a prediction probability higher than 0.5. For subject 1, the prediction probability threshold was set to 0.7 during the online acquisition, so the auxiliary file with the predictions corresponds to this threshold, and after experimenting, we re-analyzed the online data using a 0.5 threshold.

We acquired one single online block for subjects 1, 2, 5 and 7. For subject 3, we acquired three online blocks with 50%, 80%, and 85% accuracy. For subject 6, we acquired two online blocks with 80% and 85% accuracy. For subject 10, we acquired three online blocks with 55%, 70%, and 80% accuracy. It can be seen that for all subjects, the online accuracy increases with each block acquisition. The accuracy shown in Table 1 corresponds to those online blocks with the highest accuracy for each subject. For subject 4, the training and online data have poor quality (66.7% accuracy for the model and 20% accuracy for the online prediction). Subject 9 had a good model accuracy (95%) but poor-quality signals during online acquisition (50% and 20% accuracy). Post-experimental analysis of the data revealed noisy and flat signals, showing no clear pattern in the signal acquired from subjects 4 and 9, similar to the signal acquired from Subject 8 (Please see Figure 5 for the signal from patient 8). These distortions may have arisen due to the probable electrode movement, detachment, or misplacement. Thus, we decided to discard the data from Subjects 4, 8 and 9.

## 4. Discussion

It must be clear that in order to make a completely fair comparison between our system and the state-of-the-art systems, some extensive testing would be required. These tests should process the data acquired in this study with other processing pipelines, run our pipeline over the data acquired in other studies, and adapt our acquisition and processing modules to be connected to further systems found in the literature. The results obtained after this process would give us a full picture of the differences between our system and those already in place. Unfortunately, due to lack of time and materials, these tests could not be carried out.

Concentration loss and tiredness are two of the biggest challenges when it comes to EOG-based HCI. As reported in Barea et al. [43], the number of failures using this kind of system increases over time after a specific period of use. This has been seen during the development of this study, where long periods of system use have led to the appearance of irritation and watery eyes. This could be a problem for subjects who use the system for a long time. In the paper above mentioned [43], the researchers deal with this problem by retraining the system.

Another challenge related to our system is the presence of unintentional eye blinks. Eye blinks create artifacts in the EOG signal and, also, during the eye blinks, there is a slight eye movement [37]. The trials containing eye blinks can lead to a reduced model accuracy if it occurs in the training stage or to a trial misclassification if it is in the online acquisition stage. Pander et al. [44], and Merino et al. [30] have proposed methods to detect spontaneous blinks so these trials can be rejected. Yathunanthan et al. [6] proposed a system where eye blinks are automatically discarded.

Our system, like most of the available systems in the literature [19,20,21,29,30,38,43], uses a discrete approach, i.e., the user is not free to perform an action when desired, but the action must be performed at a specific time. This affects the agility of the system by increasing the time needed to perform an action. Barea et al. [38,43] and Arai et al. [25] have proposed systems with a continuous approach where the subject has no time restrictions to perform an action.

There are different ways to improve our system in future work. First, we could put in place a mechanism to detect and remove unintentional blinks. This would prevent us discarding training blocks, or could improve the training accuracy in the cases in which these unintentional blinks occur. In some cases, a continuous online classification means a considerable advantage. Therefore, it would be interesting to add the necessary strategies to perform this type of classification. Finally, by combining our system with further communication or movement assistance systems, we could check its performance in a complete HCI loop.

## 5. Conclusions

We have presented an EOG signal classification system that can achieve a 90% mean accuracy in online classifications. These results are equivalent to other state-of-the-art systems. Our system is built using only open components, showing that it is possible to avoid the usage of expensive and proprietary tools in this scope. As intended, the system is small, easy to carry, and has complete autonomy. This is achieved using OpenBCI and Raspberry Pi as hardware, connected to a power bank as a power source. 

Because of the use of open hardware and software technologies, the system is also open, easy to replicate, and can be improved or modified by someone with the required skills to build a tailored solution. The use of open technologies also helps us to obtain a cheap platform.

Finally, the resulting system is easy to connect to subsequent communication or movement assistance systems.

## Figures and Tables

**Figure 1 sensors-20-02443-f001:**
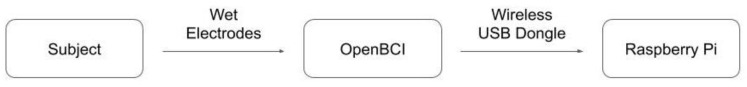
Block diagram with system connection.

**Figure 2 sensors-20-02443-f002:**
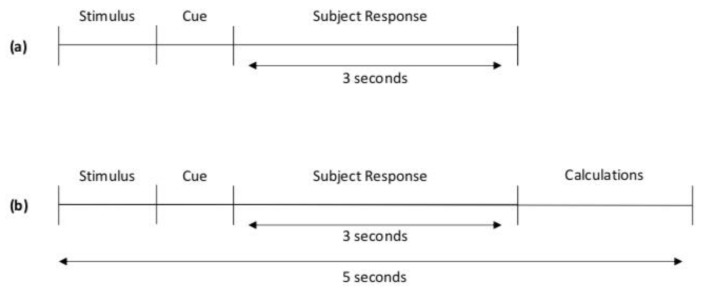
Acquisition paradigm. (**a**) Offline acquisition. (**b**) Online acquisition.

**Figure 3 sensors-20-02443-f003:**
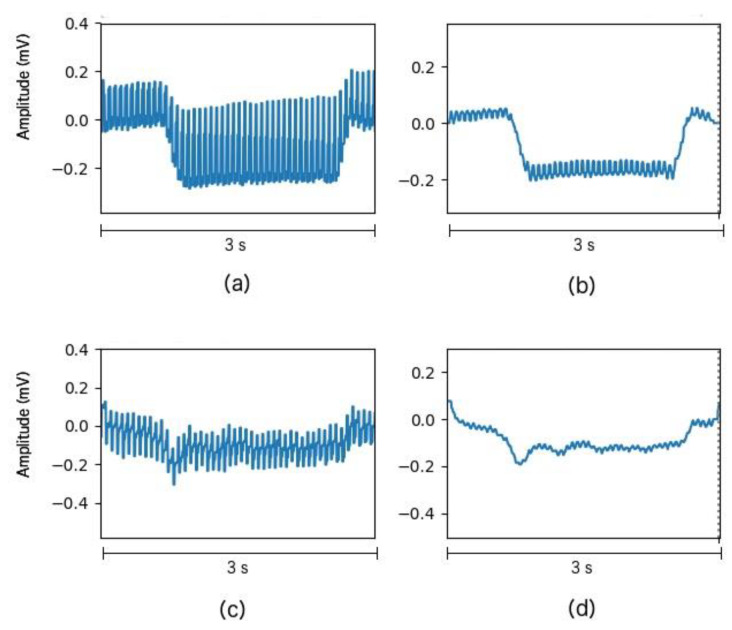
Down movement example taken from Subject 5. The *x*-axis depicts time (in seconds), and *Y*-axis represents the signal amplitude (in millivolts). (**a**) Unfiltered vertical component. (**b**) Filtered vertical component. (**c**) Unfiltered horizontal component. (**d**) Filtered horizontal component.

**Figure 4 sensors-20-02443-f004:**
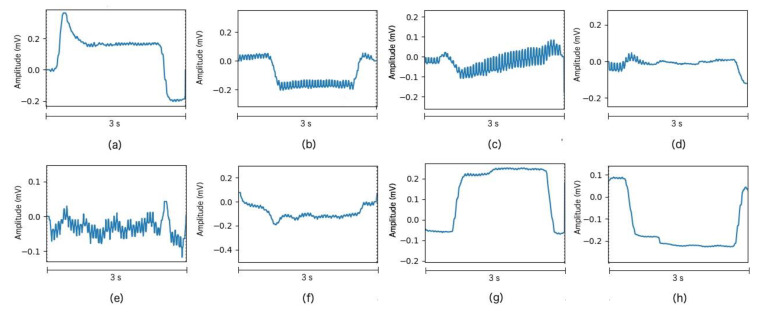
Processed signals examples taken from Subject 5. The *x*-axis depicts time (in seconds), and *Y*-axis represents the signal amplitude (in millivolts). (**a**) Vertical component for Up movement. (**b**) Vertical component for Down Movement. (**c**) Vertical component for Left movement. (**d**) Vertical component for the Right movement. (**e**) Horizontal component for Up movement. (**f**) Horizontal component for Down movement. (**g**) Horizontal component for Left movement. (**h**) Horizontal component for Right movement.

**Figure 5 sensors-20-02443-f005:**
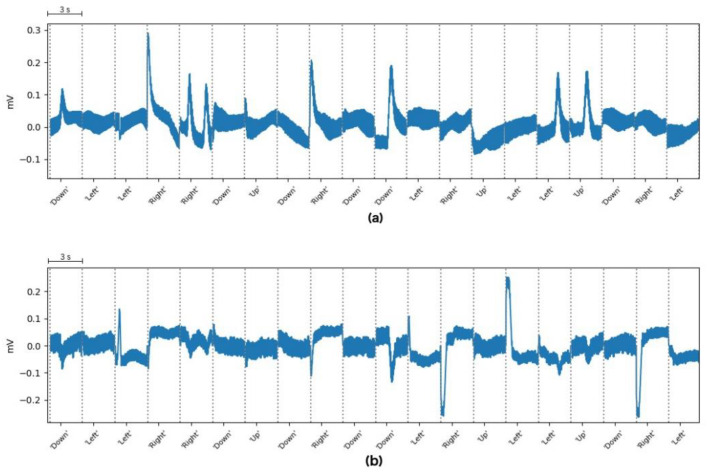
Example trials taken from Subject 8. The *x*-axis depicts time (each trial is 3 s), and *Y*-axis represents the signal amplitude (in millivolts). (**a**) Vertical component. (**b**) Horizontal component.

**Figure 6 sensors-20-02443-f006:**
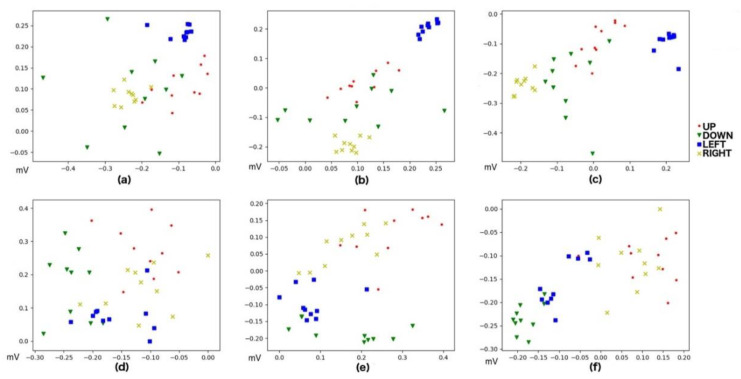
Values after Feature Extraction for Up, Down, Left, and Right movements performed by Subject 5 (100% model accuracy). Both *X*-axis and *Y*-axis depict signal values (in millivolts). (**a**) Horizontal Min vs. Max. (**b**) Horizontal Max vs. Median. (**c**) Horizontal Median vs. Min. (**d**) Vertical Min vs. Max. (**e**) Vertical Max vs. Median. (**f**) Median vs. Min.

**Figure 7 sensors-20-02443-f007:**
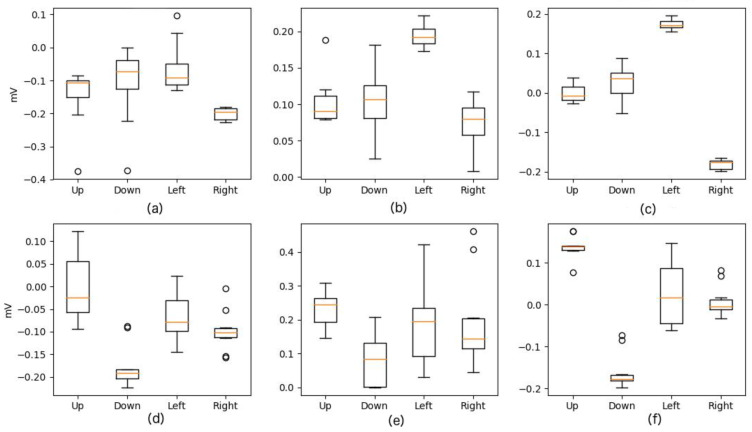
Values after Feature Extraction for Up, Down, Left, and Right Movements performed by Subject 5 (100% model accuracy). The *x*-axis depicts movement class, and *Y*-axis depicts signal amplitude (in millivolts). (**a**) Horizontal Min. (**b**) Horizontal Max. (**c**) Horizontal Median. (**d**) Vertical Min. (**e**) Vertical Max. (**f**) Vertical Median.

**Figure 8 sensors-20-02443-f008:**
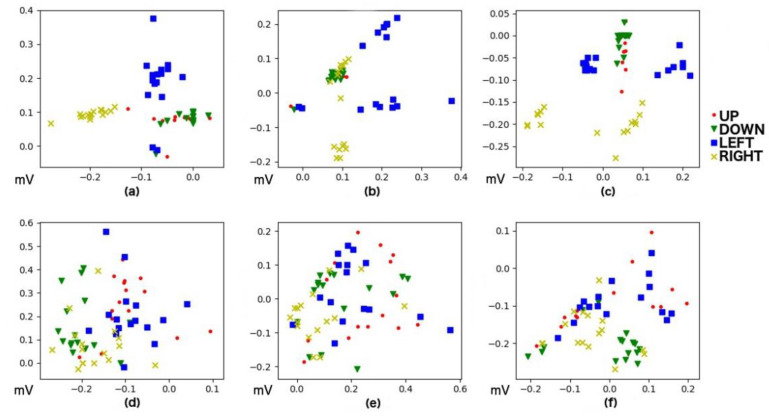
Values after Feature Extraction for Up, Down, Left, and Right movements performed by Subject 6 (78.7% model accuracy). Both *X*-axis and *Y*-axis depict signal values (in millivolts). (**a**) Horizontal Min vs. Max. (**b**) Horizontal Max vs. Median. (**c**) Horizontal Median vs. Min. (**d**) Vertical Min vs. Max. (**e**) Vertical Max vs. Median. (**f**) Median vs. Min.

**Figure 9 sensors-20-02443-f009:**
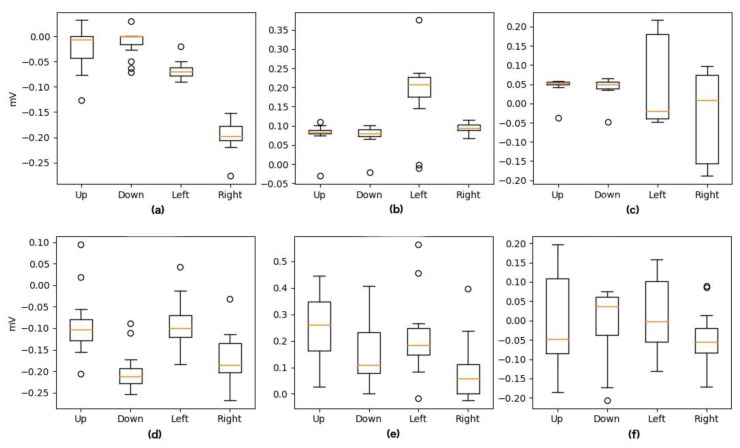
Values after Feature Extraction for Up, Down, Left, and Right Movements performed by Subject 6 (78.7% model accuracy). The *x*-axis depicts movement class, and *Y*-axis depicts signal amplitude (in millivolts). (**a**) Horizontal Min. (**b**) Horizontal Max. (**c**) Horizontal Median. (**d**) Vertical Min. (**e**) Vertical Max. (**f**) Vertical Median.

**Table 1 sensors-20-02443-t001:** Comparison of results between different studies.

Study	Movements	Acquisition	Processing	Method	Accuracy
Qi et al. [27]	Up, Down, Left, Right	Commercial	-	Offline	70%
Guo et al. [28]	Up, Down, Blink	Commercial	Laptop	Online	84%
Kherlopian et al. [24]	Left, Right, Center	Commercial	Laptop	Online	80%
Wu et al. [20]	Up, Down, Left, Right, Up-Right, Up-Left, Down-Right, Down-Left	Self-designed	Laptop	Online	88.59%
Heo et al. [26]	Up, Down, Left, Right, Blink	Self-designed	Self-designed + Laptop	Online	91.25%
Heo et al. [26]	Double Blink	Self-designed	Self-designed + Laptop	Online	95.12%
Erkaymaz et al. [29]	Up, Down, Left, Right, Blink, Tic	Commercial	Laptop	Offline	93.82%
Merino et al. [27]	Up, Down, Left, Right	Commercial	Laptop	Online	94.11%
Huang et al. [21]	Blink	Self-designed	Laptop	Online	96.7%
Lv et al. [19]	Up, Down, Left, Right	Commercial	Laptop	Offline	99%
Yathunanthan et al. [6]	Up, Down, Left, Right	Self-designed	Self-designed	Online	99%

**Table 2 sensors-20-02443-t002:** Model and Prediction Accuracies.

Subject	Model Mean Accuracy	Online Accuracy
Subject 1	100%	90%
Subject 2	100%	95%
Subject 3	92.5%	85%
Subject 5	100%	100%
Subject 6	78.7%	85%
Subject 7	97.5%	95%
Subject 10	90.8%	80%
MEAN	94.21%	90%

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
