# Peer review of "Open Software/Hardware Platform for Human-Computer Interface Based on Electrooculography (EOG) Signal Classification"

_sensors, 2020, doi:10.3390/s20092443_

Round 1

Reviewer 1 Report

A novel Open Software/Hardware platform is always welcome. However the system does not present such such novelty because is based on existing systems, and are not clearly presented new additions to it. Only is used the system hardware to capture the signal. On the software side, were used well known filters, using well-known toolboxes, also for the SVM. Again, no novelty on this side.

The paragraph related to the state-of-the-art, lines 58 to 67, is too short, does not explain correctly he work done on the references, does not place the proposed paper against previous work, does not present any technical novelty apart from the open platform usage. This paragraph should be much more extended to justify the paper by itself against the known state-of-the-art.

The feature extraction follows other research, correctly referenced in the paper. The SVM is also referenced, along the kernel used, although no clear justification for those items were done in the text. For the classification, a validation was performed, which is correct.

The testing part, with completely new data was presented as the on-line case, in section 5, with an expected decrease on accuracy. A few results, 10 subjects were presented. The discussion is in some way not very convincing "this situation is probably due to an electrode movement, detachment, or misplacement" - this fact should be noticed in the experiments performed that acquired the data. Moreover, as future work is suggested for the classification system also detect these faulty issues. As such, with the data acquired for testing further tests could be done.

Regarding the comparison to other methods, should be done with the same datasets for a fair comparison. However, from the results presented, is clear the problems of the system to obtain the best results. No discussion on this is presented.

In conclusion, there is no novelty in the techniques used, the HW and SW used. The paper is closer to a technical report than to a journal paper.

Author Response

Thank you for your review. We have prepared an extensive review response document with answers to the reviewer's questions as well as highlighting the changes in the revised manuscript.

Reviewer 2 Report

The manuscript shows the recognition results of 4 directional eye movements by using OpenBCI and Rasberry Pi. The mean accuracy with SVM was about 90% which was lower than the previous studies but acceptable in consideration with the fact that the relatively low-price devices were utilized.

The manuscript was well written and is very interesting because this study utilized the open sources only, but it should be improved to support the conclusion strongly.

My comments are as follows:

1. The reason for the low accuracy should be studied more. The signal in Fig.7 is clear enough to be recognized easily. The signals which caused the low accuracy should be found and studied more. It could be the problem of the algorithm, or it could be noise or artifacts.

2. Eye-blink (or other artifacts) problem

The low accuracy could be caused by eye blinks or other artifacts (as the authors noted in the discussion). If it is, the low accuracy might not be related to the devices or open-source codes.

I think the main contribution of this paper is that it compares the results with the open-source system to the conventional system. If the low accuracy was caused simply by the eye-blinks, it would be difficult to compare the results fairly.

3. Comparison to the conventional H/W

If it is difficult for the authors to answer the comment #2, the authors may obtain the same experiment with a conventional H/W. This experiment will prove the conclusion of the paper strongly.

Author Response

(The authors gave the same response as above.)

Round 2

Reviewer 1 Report

In this version of the paper, the authors clearly showed the limitations of the paper and showed that the purpose of the paper was to only present an open-source solution.

The authors clearly presented that and in this sense the paper is valid.

Furthers improvements were done to improve the paper as suggested by the reviewer.

The authors can also compare point out the BITALINO low-cost solution for EOG. https://plux.info/sensors/404-electrooculography-eog-sensor.html

Author Response

Response: We would like to thank the reviewer for this comment. In response to his comment, we have included a new paragraph in the Introduction section to mention this alternative device.

Changes in the manuscript: Added lines 131-134.

“It should be noted that although we have used the OpenBCI board as an acquisition system, other biosignal acquisition boards like BITalino would also have served our purpose. BITalino offers an EOG acquisition module, an open environment, and includes a Python-based API for connection and signal acquisition over Raspberry Pi.”

Reviewer 2 Report

The manuscript is well revised, and I think it is fine to be published in the Journal with a single comment.

One of the limitations of this manuscript is that the proposed system (H/W + S/W) was not compared to other existing systems or algorithms with the same dataset or algorithm (S/W should be fixed to evaluate H/W precisely, vice versa). This perspective is better to be noted in the discussion.

Author Response

Response: We agree with the reviewer and would like to thank the reviewer for this comment. In response to his comment we have included a new paragraph in the Discussion section to mention this limitation.

Changes in the manuscript: Added lines 308-316.

“It is worth mentioning that to realize a quantitative comparison between our system and other state-of-the-art systems some extensive testing would be required. These tests should process the data acquired in this study with other processing pipelines, run our pipeline over the data acquired in other studies, and adapt our acquisition and processing modules to connect it to different commercial and standard hardware systems reported in the literature. The results obtained after this process would give us detailed information on the differences between our system and those already in use. However, the software and hardware comparisons mentioned above were not performed as it was out of the scope of the current work which focused on the introduction of the here described system.”